# Crystal structure of the human PRPK–TPRKB complex

Jian Li [1,5], Xinli Ma[1,5], Surajit Banerjee[2], Hanyong Chen[1], Weiya Ma[1], Ann M. Bode[1] & Zigang Dong [3,4 ✉]

Mutations of the p53-related protein kinase (PRPK) and TP53RK-binding protein (TPRKB) cause Galloway-Mowat syndrome (GAMOS) and are found in various human cancers. We have previously shown that small compounds targeting PRPK showed anti-cancer activity against colon and skin cancer. Here we present the 2.53 Å crystal structure of the human PRPK-TPRKB-AMPPNP (adenylyl-imidodiphosphate) complex. The structure reveals details in PRPK-AMPPNP coordination and PRPK-TPRKB interaction. PRPK appears in an active conformation, albeit lacking the conventional kinase activation loop. We constructed a structural model of the human EKC/KEOPS complex, composed of PRPK, TPRKB, OSGEP, LAGE3, and GON7. Disease mutations in PRPK and TPRKB are mapped into the structure, and we show that one mutation, PRPK K238Nfs*2, lost the binding to OSGEP. Our structure also makes the virtual screening possible and paves the way for more rational drug design.

[1] The Hormel Institute, University of Minnesota, Austin, MN 55912, USA. [2] Northeastern Collaborative Access Team, Argonne National Laboratory, Argonne, IL 60439, USA. [3] College of Medicine, Zhengzhou University, Zhengzhou, China. [4] China-US (Henan) Hormel Cancer Institute, Zhengzhou, China. [5]These authors contributed equally: Jian Li, Xinli Ma. ✉email: dongzg@zzu.edu.cn

The p53-related protein kinase (PRPK, TP53RK) was initially cloned and described as a p53 interacting protein being able to phosphorylate p53 at Ser15[1]. Its binding partner, Cgi121 (TPRKB, TP53RK-binding protein), was identified by the same group through a yeast two-hybrid screen[2]. Beyond these studies, little is known about PRPK. Yet, another line of studies regarding Bud32, the yeast homologue of human PRPK, evolved rapidly. A complex referred to as EKC/KEOPS (stands for endopeptidase-like kinase chromatin-associated/kinase, endopeptidase, and other proteins of small size) was identified as a telomere regulator[3] and a transcription complex[4]. This complex is composed of Bud32 (piD261, YGR262C), Cgi121 (YML036W), Kae1 (Ykr038c), Gon7 (Yjl184w), and Pcc1 (YKR095W). The EKC/KEOPS complex was later found to be essential for a universal tRNA modification, threonyl-carbamoyl adenosine (t6A), found in all tRNAs that pair with ANN codons. This modification strengthens the A–U codon–anticodon interaction on the ribosome[5]. Kae1, being an extremely well-conserved protein (TsaD/YgjD as the *E. coli* ortholog), is the catalytic subunit and transfers the L-threonyl-carbamoyl moiety to tRNA. Kae1, Bud32, and Pcc1 may be the minimum set required for t6A modification, while the addition of Cgi121 confers maximal activity. In the KEOPS complex, Kae1 switches the kinase activity of Bud32 to ATPase activity. Bud32 is responsible for the ATPase activity of the KEOPS complex and ATPase activity is required for t6a synthesis. Within the complex, Kae1 and Pcc1 form the tRNA binding core[6].

The human version of the KEOPS complex was finalized with the identification of C14ORF142 as the Gon7 ortholog in human. Thus, the human KEOPS complex is composed of OSGEP (Kae1 in yeast), PRPK (Bud32 in yeast), TPRKB (Cgi121 in yeast), LAGE3 (Pcc1 in yeast), and GON7[7]. Surprisingly, human KEOPS complex mutations lead to Galloway–Mowat syndrome (GAMOS), a rare autosomal recessive disease characterized by early onset nephrotic syndrome and microcephaly[8–10]. Knockdown of OSGEP, PRPK, or TPRKB inhibits cell proliferation, impairs protein translation, activates DNA damage response signaling, and reduces cell migration of human podocytes[9].

PRPK was first shown to be phosphorylated at Ser250 and thus activated by protein kinase B (PKB/Akt)[11]. We further demonstrated that T-LAK cell-originated protein kinase (TOPK, PBK) phosphorylates PRPK at Ser250[12]. Metastatic human colon adenocarcinomas and human cutaneous squamous cell carcinoma samples all display higher levels of Ser250 phosphorylated PRPK compared with earlier stages of colon adenocarcinomas and normal skin, respectively. Both the PRPK protein and Ser250 phosphorylation are critical for colon cancer metastasis and skin carcinogenesis in mouse models. Importantly, small molecules targeting PRPK showed promising efficacy in both models[12–14]. Despite a central role in the synthesis of an essential tRNA modification, the elevated PRPK protein level, S250 phosphorylation level, and activity level in metastatic cancer compared to normal tissues could be preferentially targeted and toxicity to healthy cells may be minimized. In our studies, mice have been benefited from the treatment using PRPK inhibitors[13,14]. Similar to our discovery, PRPK is also highly expressed and shown as a valuable target in multiple myeloma[15].

As a very promising drug target, its structure will certainly facilitate virtual screening and more rational drug development. In this study, we report the 2.53 Å crystal structure of the human PRPK–TPRKB–AMPPNP complex.

## Results

**Overall structure of the human PRPK–TPRKB–AMPPNP ternary complex.** The human kinase superfamily (Kinome) has been grouped into eukaryotic protein kinases (ePKs) and, due to lack of sequence similarity, atypical protein kinases (aPKs)[16]. PRPK belongs to the ePK group labeled as "Other". This group has certain conserved elements in their kinase domain but cannot be assigned into the major ePK groups[17]. Within this group, PRPK is placed in the Bud32 family, which is an ancient family with one member found in almost all eukaryotic and archaeal genomes.

Here we describe the crystal structure of human PRPK–TPRKB bound to AMPPNP (adenylyl-imidodiphosphate) at a resolution of 2.53 Å. PRPK roughly adopts a kinase domain that splits into the N-lobe and the C-lobe. The ATP analogue, AMPPNP, is bound in the cleft between the N- and C-lobe. The C-lobe is considerably smaller than a typical kinase (Fig. 1; Supplementary Fig. S1a). The TPRKB protein is comprised of a central four-stranded antiparallel β-sheet flanked by two and seven α-helices on either side (Fig. 1; Supplementary Fig. S2). TPRKB uses helices α2, α8, and α9 from one side of the β-sheet

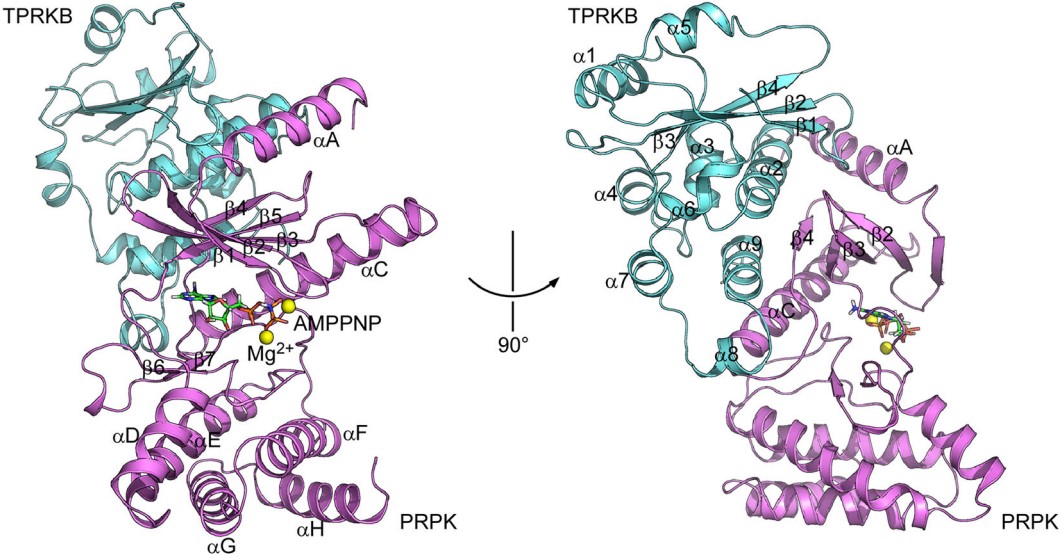

**Fig. 1 Overall structure of the human PRPK–TPRKB–AMPPNP complex.** PRPK adopts a kinase fold with a briefed C-lobe. TPRKB uses the β1-β2 loop, α2, α8, and α9 to interact with the PRPK N-lobe. PRPK is colored in violet and TPRKB in aquamarine. AMPPNP is shown as stick and Mg$^{2+}$ as yellow spheres.

and the loop between β1-β2 to interact with the N-lobe of PRPK (Fig. 1). The overall structure is similar to the archaeal and yeast Bud31–Cgi121 structures[18,19], but with unique features described below.

**Conserved elements in the PRPK kinase domain.** Analogous to the prototype protein kinase A (PKA), human PRPK has several conserved features. Between the β-strands β1 and β2 lie a conserved glycine-rich ATP-binding loop (also known as G-loop or P-loop) with the typical motif GxGxxG. The first glycine is present in ~95% of all kinases, the second in more than 99% of kinases, and the third is conserved in ~85% of kinases. The third glycine is substituted preferentially with small amino acids such as alanine or serine[20,21]. The G-loop in PRPK is composed of $^{40}$KQGAEA$^{45}$ (Fig. 2a; Supplementary Fig. S1a). We infer that PRPK G42 corresponds to the most conserved second glycine of the consensus motif. G42 -NH forms a hydrogen bond with the A45 carbonyl oxygen, similar to the hydrogen bond between the second and third glycine in the PKA G-loop. The PRPK G-loop does not fold over to directly contact the AMPPNP γ-phosphate (~5 Å), suggesting a status not yet ready for catalysis. In PRPK, the third glycine of the G-loop motif is replaced with an alanine. Strikingly, in mammalian and zebrafish PRPK, the position of the first glycine in this motif is occupied by a lysine or glutamine, two amino acids with large sidechains. In archaea *Methanocaldococcus jannaschii* (*M. jannaschii*) this position is a standard glycine and in yeast it is a serine with a small sidechain (Supplementary Fig. S1a). Mutations localized to glycine residues in this consensus motif are known to typically disrupt the G-loop conformation and/or sterically interfere with ATP binding and are poorly tolerated[20]. Residue K40 in human PRPK protrudes downwards from the G-loop and appears like a hindrance to the ATP binding pocket (Fig. 2c). Indeed, K40A mutation slightly increased the autophosphorylation activity of the PRPK–TPRKB complex (Fig. 2d). Thus, different types of amino acids at the position of the first glycine may reflect different activities or regulatory mechanisms of PRPK from various species. Unexpectedly, substitution of the most conserved second glycine by alanine (G42A) in the PRPK–TPRKB complex greatly stimulated its autophosphorylation activity (Fig. 2d). We could only find that in the proto-oncogene B-Raf, replacement of the third glycine with alanine shows a similar stimulatory effect[22].

In PKA, an invariant lysine (K72) of the strand β3 holds the α- and β-phosphates in position. A nearly invariant glutamate (E91) of helix αC forms a salt bridge with the invariant K72 of β3, stabilizing its interaction with the α- and β-phosphates. The presence of this salt bridge is a prerequisite for the formation of active protein kinases[21]. These features are well preserved in our PRPK structure, suggesting an active conformation. Specifically, invariant lysine (K60) of β3 makes direct contact with α-phosphate of AMPPNP (~2.6 Å), and E84 of helix αC forms a salt bridge with K60 (~2.8 Å; Fig. 2b). As expected, the K60A mutation almost abolished the autophosphorylation activity (Fig. 2d).

The C-lobe contains a conserved catalytic loop with the motif HxDxxxxN, where D is the catalytic base that accepts the hydrogen removed from the hydroxyl group being phosphorylated. In PRPK, this motif is $^{160}$HGDLTTSN$^{167}$ and D162 is the catalytic residue (Fig. 2a, b; Supplementary Fig. S1a). The last asparagine in this motif (N167) coordinates one of the Mg$^{2+}$ ions used during catalysis (distance to Mg$^{2+}$ ~2.4 Å), similar to PKA (Fig. 2b). Expectedly, the D162N mutant had dramatically reduced autophosphorylation activity (Fig. 2d).

Another conserved feature of the C-lobe is the DFG loop. The aspartate in this motif chelates one of the Mg$^{2+}$ ions that bridges

the β- and γ-phosphates of ATP and positions the γ-phosphate for transfer to the substrate[21]. In PRPK, the exact $^{183}$DFG$^{185}$ plays the same role, with the D183 sidechain carboxy oxygens in close proximity to both Mg$^{2+}$ ions (within 2.8 Å; Fig. 2a, b; Supplementary Fig. S1a). Consistent with a critical role, the D183A mutation completely abolished the autophosphorylation activity (Fig. 2d).

In PKA, the DFG loop precedes the activation loop, which contains a phosphorylatable residue and its phosphorylation is usually required for enzyme activation. The activation loop ends with an APE (Ala–Pro–Glu) segment, which anchors the activation loop to the C-lobe. Between the phosphorylated residue and the APE motif lies the P+1 loop, which interacts with the residue adjacent to the phosphorylated residue of the peptide substrate. Although not all kinase require phosphorylation of an activation segment residue to became active, the presence of an activation loop is a typical kinase configuration[21]. Surprisingly, the activation loop is completely absent in PRPK, substituted by a 7 amino acid linker between the DFG loop and helix αF (Fig. 2a; Supplementary Fig. S1a). Helix αF is the last structural element that has an equivalent in PKA. Helix αF is a very hydrophobic helix buried inside PKA and the entire C-lobe is organized around this helix. In PRPK, the helix αF is exposed in one side and the C-lobe is further capped with helices αG–αH that follow helix αF (Fig. 2a; Supplementary Fig. S1a). Fewer structural elements after helix αF and the absence of the activation loop account for the smaller C-lobe in PRPK compared with PKA.

Nearly all active kinases contain this aforementioned K/E/D/D signature motif (K60, E84, D162, and D183 in PRPK) that plays important structural and catalytic roles[21]. These residues are extremely conserved at the primary sequence level (Supplementary Fig. S1a), at the structural level (Supplementary Fig. S1b), and at the functional level (Fig. 2d)[18] in human, yeast, and archaeal Bud32. Serine/threonine-protein kinase Rio2 is one of the top structural homologs identified by the Dali server (PDB ID 4GYI, Z score 13.6, rmsd 3.3 Å, for 190 structurally aligned residues)[23]. Rio2, as an atypical kinase, also lacks the activation loop and the last two helices of the canonical ePK C-lobe. Both PRPK and Rio2 have been shown to have in vitro autophosphorylation activity and ATPase activity[6,18,24]. Lastly, PRPK has a unique extension, helix αA, preceding strand β1. This helix is involved in the TPRKB interaction, and adopts a completely different orientation compared to the yeast Bud32–Cgi121 structure (Fig. 1; Supplementary Fig. S3a).

**Coordination of the AMPPNP.** Like a typical kinase, AMPPNP is well coordinated in the ATP binding pocket of PRPK. Between the adenine base of AMPPNP and the backbone of the kinase hinge region, two key hydrogen bonds are formed. Specifically, the 6-amino group of the adenine base forms a hydrogen bond with the carbonyl oxygen of E114 (~3.2 Å). The N-1 of the adenine ring forms a hydrogen bond with the main chain -NH group of the I116 hinge residue (~3.2 Å; Fig. 2c). The adenine base interacts with several hydrophobic residues in the ATP-binding pocket including V47, V58, and M113 from the N-lobe, and L169 and I182 from the C-lobe (Fig. 2c). Finally, the phosphate groups of AMPPNP and N167 and D183 sidechains are bridged by two Mg$^{2+}$ ions (Fig. 2b). The ATP binding mechanisms are conserved among the human, yeast, and archaeal Bud32 (Supplementary Fig. S1a).

**The catalytic spine and regulatory spine.** Two hydrophobic "spines" are important for the structure of the active conformation of protein kinases. They are composed of amino acid

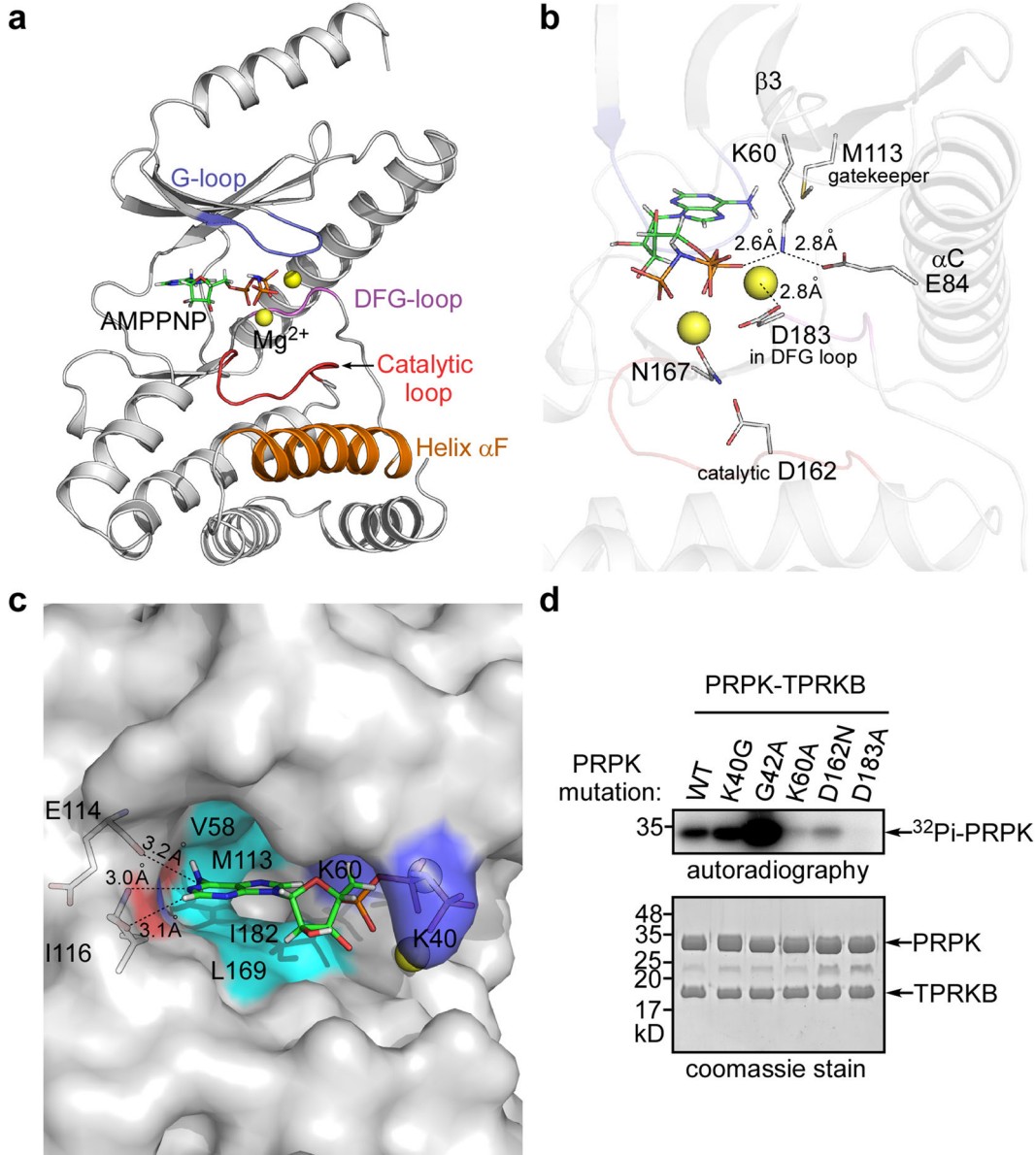

**Fig. 2 The PRPK ATP binding pocket is in an active conformation. a** Conserved kinase elements are color coded in the PRPK structure. PRPK lacks the conventional kinase activation loop between the DFG loop and helix αF. In addition, PRPK does not have the elaborated helices after αF, seen in conventional kinases. **b** Close up view of the PRPK ATP binding pocket. The invariant K60 of β3 holds the α-phosphate of AMPPNP. K60 itself is stabilized by a salt bridge with E84 of helix αC. Two $Mg^{2+}$ ions are coordinated by D183 of the DFG loop and N167 of the catalytic loop, respectively. D162 is the catalytic residue, and M113 is the gatekeeper residue. This configuration satisfies the requirement for an active kinase. Charged interactions are labeled with dashed lines and the distances are indicated. **c** Close up view of the PRPK ATP binding pocket in surface representation. E114 and I116 from the hinge region form hydrogen bonds with one edge of the adenine ring. The adenine base is surrounded by hydrophobic resides V47, V58, M113, L169, and I182 (V47 is on the roof of the pocket and could not be seen from this angle). Notably, the first glycine of the GxGxxG G-loop motif is replaced with K40 in human PRPK. The surface area of K40 is colored blue, and it may constitute a hindrance to ATP binding. Hydrogen bonds are labeled with dashed lines and the distances are indicated. **d** In vitro kinase assay showing the autophosphorylation activity of the wild type and mutant PRPK–TPRKB complexes. Uncropped images of gel and autoradiograph are shown in Supplementary Fig. S6.

residues that are non-contiguous in the primary structure. Both spines are assembled in the active conformation and disorganized in inactive conformations[25]. In PRPK, the catalytic spine comprises residues V47, V58, adenine ring of AMPPNP, L169, L170, M168, V122, L203, and F207, and it is directly anchored to the carboxyl end of helix αF (Fig. 3a). The regulatory spine contains residues V101, L88, F184, and H160, and it is anchored to helix αF by a hydrogen bond between an invariant aspartate D199 in helix αF and the backbone nitrogen of H160 (~2.8 Å; Fig. 3b).

F184 of the DFG loop reaches into a pocket formed by L88 and H160, adopting a so-called DFG-in conformation, which helps maintain the D183 sidechain in a position capable of coordinating magnesium. Collectively, both spines in PRPK are well assembled, suggesting an active conformation. Indeed, the PRPK–TPRKB complex has autophosphorylation activity (Fig. 2d). However, PRPK itself only displayed extremely low autophosphorylation activity (Fig. 3c, d), and this activity could be strongly stimulated by TPRKB, similar as seen in the archaeal

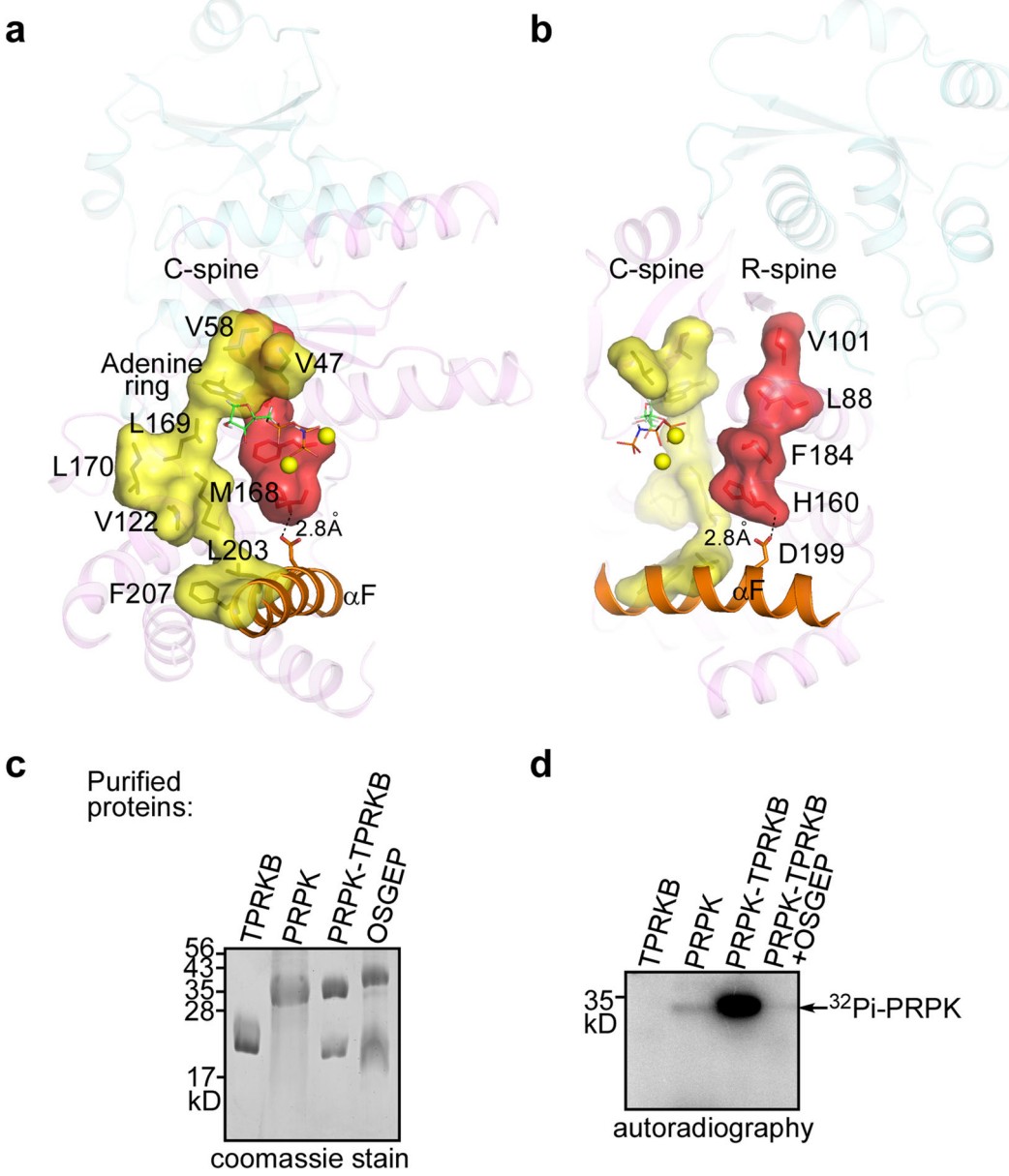

**Fig. 3 Two spines of PRPK. a** The catalytic spine (C-spine) is formed by V47, V58, adenine ring of AMPPNP, L169, L170, M168, V122, L203, and F207 (from top to bottom), and it is directly anchored to the carboxyl end of the helix αF. The C-spine is well assembled. **b** The regulatory spine (R-spine) is formed by V101, L88, F184, and H160, and it is anchored to helix αF by a hydrogen bond between the D199 carboxyl O atom and –NH group of H160 (dashed line). The R-spine is also well organized and F184 of the DFG loop adopts an active DFG-in conformation. **c** Purified proteins used in the kinase assay. **d** In vitro kinase assay showing the regulation of PRPK autophosphorylation activity by TPRKB and OSGEP. Uncropped images of gel and autoradiograph are shown in Supplementary Fig. S6.

and yeast proteins[18,26]. Mechanistically, PRPK N-lobe makes extensive interactions with TPRKB. Particularly, PRPK residue V101 from the regulatory spine forms a hydrogen bond with TPRKB S170 (Fig. 4). Thus, TPRKB may help maintain the regulatory spine in an assembled states and the PRPK in an active conformation. One possibility is that TPRKB regulates PRPK ATP binding. Another possibility is that TPRKB does not influence PRPK ATP binding, but help position the catalytic elements ready for catalysis.

Located between the two hydrophobic spines is the gatekeeper residue deep in the ATP-binding pocket[25]. The size of the gatekeeper residue determines the size of the binding pocket, and it is thus a gatekeeper for which nucleotides, ATP analogs, and inhibitors can bind[27]. The gatekeeper residue in PRPK

is a methionine (M113), the same as for PKA (Fig. 2b, c; Supplementary Fig. S1a).

**Interaction with TPRKB in the complex.** The interface between PRPK and TPRKB, calculated by PISA, is 1423 Å$^2$ [28]. The contact surfaces are comprised of conserved residues from both molecules (Fig. 4a, b). For PRPK, the interface is composed of β4, a loop between β2-β3, and helices αA and αC, all of which are from the N-lobe (Fig. 1; Fig. 4c). For TPRKB, the interface is centered on helices α2 and α9 and flanked on either side by a loop between β1-β2 and by helix α8 (Fig. 1; Fig. 4c). Notably, helix α8 is a segment of loop in the TPRKB individual structure. Upon complex formation, this loop moves towards PRPK and reorganizes

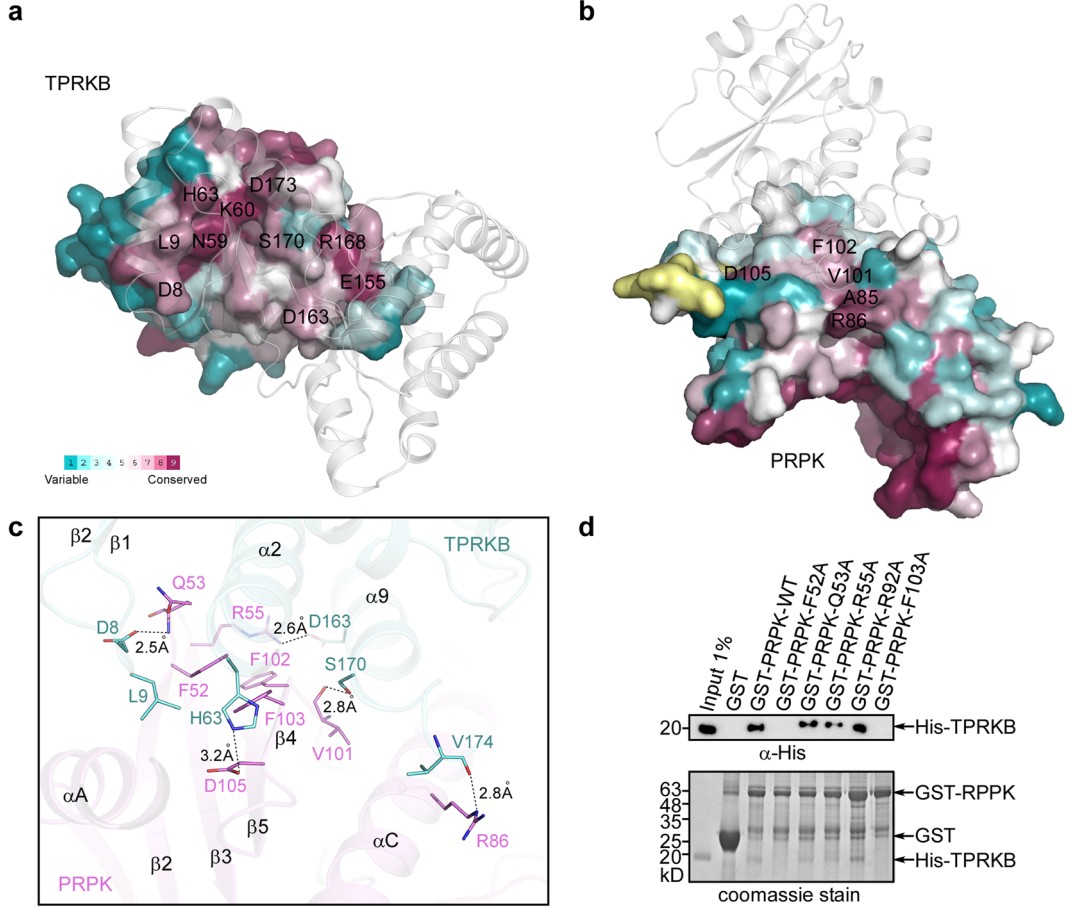

**Fig. 4 Interaction between PRPK and TPRKB. a** ConSurf style representation of the TPRKB surface involved in PRPK interaction. PRPK is shown as semi-transparent cartoon in white color. **b** ConSurf style representation of the PRPK surface involved in TPRKB interaction. TPRKB is shown as semi-transparent cartoon in white color. **c** Detailed interaction between PRPK and TPRKB. PRPK F52, F102, and F103 constitute a hydrophobic surface hosting the TPRKB helix α2. Hydrogen bonds are formed between PRPK Q53 and TPRKB D8, between PRPK R86 and TPRKB V174, and lastly between PRPK V101 and TPRKB S170 (dashed lines). Salt bridges link PRPK R55 with TPRKB D163, and link PRPK D105 with TPRKB H63 (dashed lines). **d** Pulldown of His-tagged TPRKB by GST-tagged PRPK mutants. GST-tagged PRPK was first immobilized on glutathione Sepharose beads and then incubated with a purified His-tagged TPRKB protein. After extensive washing, His-TPRKB bound to PRPK was detected using a His antibody. A representative result from at least three repetitions is shown. Uncropped images of gel and blot are shown in Supplementary Fig. S6.

into a helix. No other major structural change could be observed (Supplementary Fig. S3b). This is different from yeast protein, where major structural change occurs in a segment between helix α1 and strand β3. This region is a short helix in the apo Cgi121 structure, but transforms into a loop in the Bud32–Cgi121 complex structure (Supplementary Fig. S3c).

In detail, a combination of hydrogen bonding, electrostatic and hydrophobic interactions contribute to the binding of two molecules. The PRPK Q53 main chain –NH forms a hydrogen bond with the TPRKB D8 side-chain oxygen atom (~2.5 Å). The PRPK V101 carbonyl O atom forms a hydrogen bond with the TPRKB S170 -OH group (~2.8 Å) and last, the PRPK R86 NE atom forms a hydrogen bond with the TPRKB V174 carbonyl O atom (~2.8 Å). Of the salt bridges, PRPK R55 interacts with TPRKB D163 (~2.6 Å) and PRPK D105 interacts with TPRKB H63 (~3.2 Å). Of hydrophobic interactions, PRPK F52, F102, and F103 form a hydrophobic patch on which sits the TPRKB helix α2. Meanwhile, TPRKB L9 at the tip of the β1-β2 loop, reaches into a pocket formed by PRPK residues from helix αA and β-sheet β2-β5 (Fig. 4c). The aforementioned residues account for most of the conserved amino acids seen at the contact surfaces (Fig. 4a, b). Exceptions are TPRKB N59/K60 of helix α2, which contact the PRPK β4. Also, the TPRKB E155 of helix α8 and R168

of helix α9 form an intra-molecular salt bridge and contact the carboxyl end of the PRPK helix αC.

Furthermore, we selected a panel of PRPK mutations, and examined their binding to TPRKB through in vitro GST pulldown assays. Of these, mutation of two hydrophobic resides, F52A and F103A had the most dramatic effect, nearly abolishing the interaction (Fig. 4d). Collectively, we have identified two residues from PRPK, F52, and F103, that are critical for the TPRKB interaction.

**Identification of methotrexate as a PRPK inhibitor through in silico screening.** With the PRPK crystal structure available, we performed virtual screening of the FDA approved drugs to identify PRPK inhibitors. Methotrexate (MTX) was identified as the second-best candidate behind ATP. Based on the GlideScore (GScore) of the docking outputs, MTX ranked higher compared with the three previously identified inhibitors[13,14]. In the docking model, MTX fits nicely into the ATP binding pocket of PRPK. Several hydrogen bonds are formed between the MTX pteridine moiety and the PRPK hinge region, and between the MTX glutamate moiety and the PRPK G-loop (Fig. 5a, b). In vitro pulldown using Methotrexate-agarose demonstrated that MTX binds

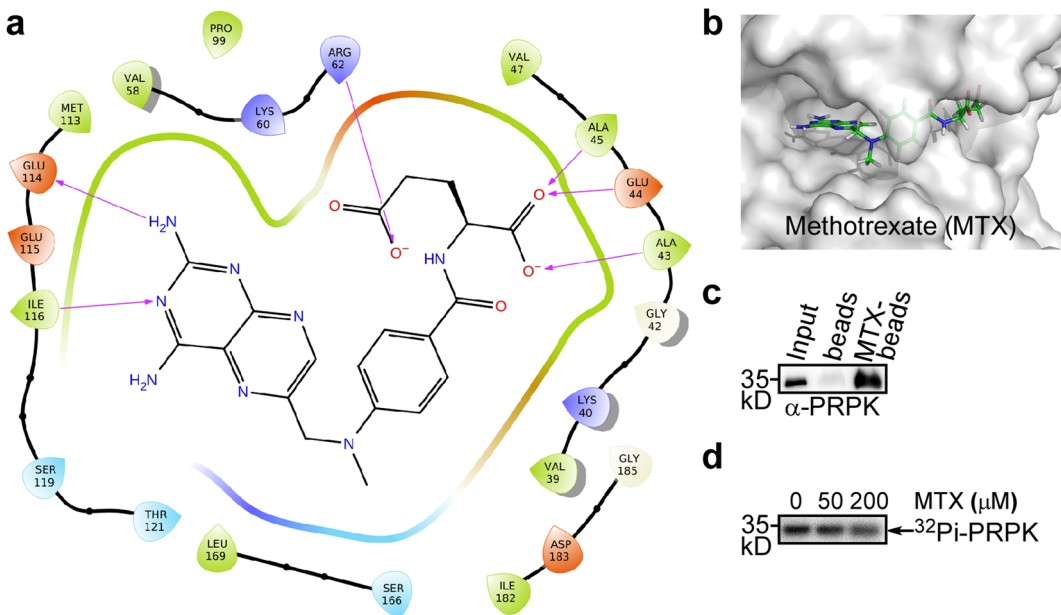

**Fig. 5 Identification of methotrexate as a PRPK inhibitor. a** Detailed MTX binding interactions with PRPK. **b** MTX fits nicely to the PRPK ATP binding pocket. **c** The binding of MTX to PRPK was confirmed by in vitro pulldown assay using MTX-agarose. **d** Inhibition of PRPK–TPRKB autophosphorylation activity by MTX. Uncropped images of blot and autoradiograph are shown in Supplementary Fig. S6.

to PRPK (Fig. 5c). MTX also showed inhibition of the PRPK–TPRKB autophosphorylation activity (Fig. 5d). Collectively, this demonstrates that virtual screening based on our PRPK–TPRKB crystal structure is very useful for lead identification.

**Model of the PRPK–TPRKB–OSGEP–LAGE3–GON7 complex**. In *M. jannaschii*, PRPK and OSGEP homologues are fused into a single protein, MJ1130. The *M. jannaschii* MJ1130 Kae1 domain (mjKae1) has approximately 50% identity with human OSGEP, and the Bud32 domain (mjBud32) has 36% identity with human PRPK. By aligning our PRPK–TPRKB structure and the published OSGEP–LAGE3–GON7 structure to the 2.7 Å mjBud32–Kae1 fusion protein structure, we have built a model of the 5-protein complex (Fig. 6a). No obvious clashes are observed between PRPK and OSGEP. Notably, the PRPK surface involved in the OSGEP interaction is much more conserved than the TPRKB binding surface (Supplementary Fig. S4a; Fig. 4b). As expected, the PRPK ATP binding pocket is also highly conserved (Supplementary Fig. S4a). Upon complexation with OSGEP, the conventional substrate binding site of a kinase was completely blocked (Supplementary Fig. S4b). Indeed, upon addition of OSGEP, PRPK–TPRKB autophosphorylation activity was diminished (Fig. 3d). This is in line with the phenomenon found in yeast where Kae1 inhibits the kinase activity of Bud32[26] and switches the kinase activity of Bud32 to ATPase activity[6].

The regions that PRPK uses to interact with OSGEP include the loop between β3-αC, helix αC, the loop between β1-β2, a portion of the catalytic loop, helix αF, and the C-terminus of helix αH (Fig. 6b). The PRPK helix αH and the C-terminal tail are placed close to the OSGEP catalytic center and a potential regulatory mechanism will be discussed later. Through analysis of the inter-domain interactions within the mjKae1–Bud32 fusion protein structure, we focused our investigation on three pairs of salt bridges. The involved residues are conserved in human PRPK and OSGEP and are in proximity in our model. Thus, the three salt bridges may exist in the actual PRPK–OSGEP complex. For PRPK, these residues are K65, R80, and K205. To investigate this idea, we performed in vitro GST pulldown assays. As predicted,

the PRPK K65A, R80A, and K205A mutants all have reduced binding towards OSGEP (Fig. 6c). This suggests that the β3-αC loop, helix αC, and αF are important for the OSGEP interaction and our aligned structural model is valid.

Further expanding this analysis, we included a PRPK K238Nfs*2 mutant found in several cancer cell lines and patient samples (Table 1), and also included S250A, and S250E mutants implicated in the oncogenic function of PRPK[12–14]. Of these, K238Nfs*2 mutation nearly abolished the interaction with OSGEP, whereas the S250A, and S250E mutations had a neglectable effect (Fig. 6c).

**Structure-based analysis of the disease mutations**. PRPK is frequently mutated in human Galloway–Mowat syndrome and in various cancers. We have analyzed most of these recurring mutations from a structural view and summarized our findings in Tables 1, 2. Of these, we have confirmed that the K238Nfs*2 mutation affects OSGEP binding (Fig. 6c). This mutation causes the K238 PRPK residue to change into an asparagine and residues 239 to C-terminus are deleted. Based on the homologous mjKae1–Bud32 structure and our model, the PRPK C-terminal tail directly contacts OSGEP. Previous studies in archaea and yeast systems have shown that deletion of the C-terminal tail has no impact on the PRPK–OSGEP interaction[18]. However, the K238Nfs*2 mutation has a longer deletion and loses half of the helix αH (Fig. 6b). In our structure, a salt bridge links K238 of helix αH and E194 of helix αF (~3.0 Å) and helix αE-H forms a compact 4-helix bundle. Because the PRPK helix αF is important for the OSGEP interaction, we propose that, by disrupting the salt bridge and helix bundle, the K238Nfs*2 mutation may destabilize the C-lobe helices αH, αF, and thus abolish OSGEP binding. In yeast, Bud32 is known to be required for Kae1 catalytic activity[6]. Therefore, the PRPK K238Nfs*2 mutation also could affect OSGEP enzyme activity.

**Discussion**
In this study, we report the crystal structure of the human PRPK–TPRKB complex bound to AMPPNP. Our structure

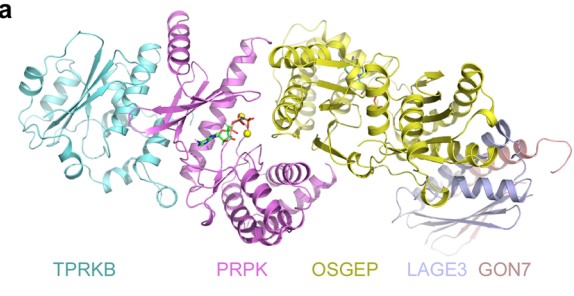

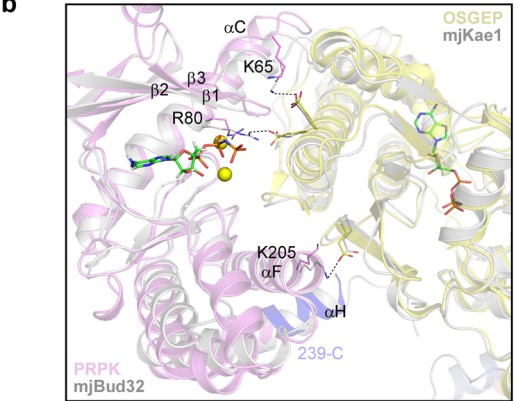

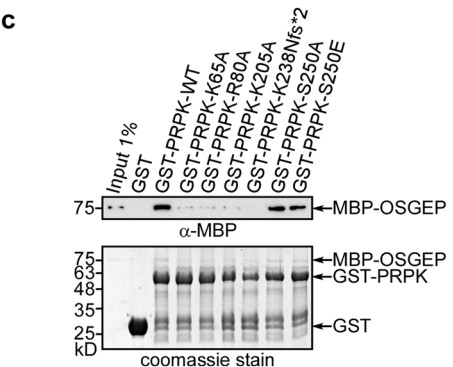

**Fig. 6 A model of the human EKC/KEOPS complex. a** The human EKC/ KEOPS complex structural model, composed of TPRKB, PRPK, OSGEP, LAGE3, and GON7, is constructed by aligning our structure and the OSGEP–LAGE3–GON7 structure (PDB ID 6GWJ) to the mjBud32–Kae1 structure (PDB ID 3EN9). AMPPNP is shown as stick and Mg$^{2+}$ as yellow spheres. **b** Detailed view of the interface between PRPK (violet) and OSGEP (yellow) in our model, aligned to the mjBud32–Kae1 crystal structure (gray). Three salt bridges found in mjBud32–Kae1 appear conserved at the PRPK–OSGEP interface. The involved residues in PRPK are K65, R80, and K205. The region deleted in the PRPK K238Nfs*2 mutant (residues 239-C) is colored light blue. **c** Pulldown of MBP-tagged OSGEP by GST-tagged PRPK mutants. GST-tagged PRPK was first immobilized on glutathione Sepharose beads and then incubated with a purified MBP-tagged OSGEP protein. After extensive washing, MBP-OSGEP bound to PRPK was detected using an MBP antibody. A representative result from at least 3 repetitions is shown. Uncropped images of gel and blot are shown in Supplementary Fig. S6.

reveals a detailed PRPK-ligand interaction and a PRPK–TPRKB inter-molecular interaction. PRPK appears in an active conformation, except that it lacks the canonical kinase activation loop. Based on our PRPK–TPRKB–OSGEP–LAGE3–GON7 model, the PRPK substrate binding site is occluded and thus may not function as a kinase in this complex. However, a PRPK–TPRKB dimer could also possibly exists in cells and could

phosphorylate substrates. In previous studies, we identified fusidic acid, rocuronium bromide, and betamethasone 17-valerate as PRPK inhibitors. These inhibitors are effective in preventing colon cancer metastasis and skin carcinogenesis[13,14]. Here, through structure-based virtual screening, we find MTX could bind and inhibit PRPK activity. Although the specificity of MTX and its efficacy in mouse models requires further studies, this demonstrates the value of virtual screening using our PRPK–TPRKB crystal structure. MTX is wildly used for chemotherapy and the treatment of rheumatoid arthritis (RA). Multiple mechanisms could be utilized by MTX[29]. Recently, MTX has been identified as an inhibitor of the JAK/STAT pathway and may potentially be acting directly as a JAK kinase inhibitor[30,31]. This study and ours suggest that MTX could be further exploited as a kinase inhibitor.

We previously showed that PRPK is phosphorylated by TOPK at Ser250. Wild-type PRPK, but not a PRPK S250A mutant, promotes colon cancer metastasis in a mouse model[12]. In our PRPK/TPRKB structure, R245 is the last residue that could be confidently modeled. Nevertheless, in our PRPK–TPRKB–OSGEP–LAGE3–GON7 complex model, the last helix αH of PRPK is next to and oriented towards the OSGEP catalytic center (Fig. 6b; Supplementary Fig. S5b). We propose that the PRPK C-terminal tail may play a regulatory role in OSGEP activity, possibly inhibitory, and Ser250 phosphorylation may relieve the inhibition. The reason is as follows:

(a) PRPK helix αH has some conserved residues and the C-terminal tail is extremely conserved, including a nearly invariant RGR motif (PRPK residues 245–247; Supplementary Fig. S5a). Some conserved residues of helix αH could be readily explained. Both in our PRPK–OSGEP model and in the mjBud32–Kae1 crystal structure, 4 hydrophobic residues from helix αH form hydrophobic interactions with helices αF-αG. In both our model and the crystal structure, these 4 residues align well in space (Supplementary Fig. S1a; Supplementary Fig. S5a, b). This suggests that in the PRPK–OSGEP structure, PRPK helix αH and the C-terminal tail may follow a similar trajectory as the corresponding part of mjBud32 in the mjBud32–Kae1 crystal structure, including the RGR motif. In mjBud32–Kae1, the 3 amino acids are RAR, with the second arginine reaching into the Kae1 catalytic pocket. We propose that the RGR motif in PRPK also adopts a similar configuration. This is because the OSGEP/Kae1 residues surrounding the two arginines are conserved and also because the Cα positions of the first arginine are very closely aligned in our PRPK–OSGEP model and in the mjBud32–Kae1 crystal structure. By pointing to the OSGEP catalytic center, as in the case of the mjBud32–Kae1 crystal structure, the second arginine of the PRPK RGR motif may interfere with substrate binding and plays an inhibitory role. Continuing to follow the path of the mjBud32 tail in the crystal structure, this potentially places PRPK Ser250 close to, and PRPK M251–V252 right on top of a hydrophobic surface on OSGEP (Supplementary Fig. S5b). Phosphorylation of Ser250 may destabilize these interactions and relieve the inhibition posed by the second arginine of the RGR motif. Interestingly, serine and threonine, two phosphorylatable amino acids that frequently occupy the Ser250 position, are immediately followed by phosphomimetics glutamic acid and aspartic acid in the occurrence frequency (Supplementary Fig. S5a). Glutamic acid or aspartic acid take this position in nematodes, such as in the model organism *C. elegans*, and in several kinds of fungi. Perhaps, these species lack such a regulatory mechanism. Our hypothesis is consistent with the fact that knockdown of OSGEP and PRPK inhibits cell proliferation and reduces cell migration[9], while phosphorylated PRPK Ser250 plays an oncogenic and S250A mutant plays an inhibitory role in metastasis[12–14].

**Table 1 Disease mutations of human PRPK.**

| PRPK Mutation | Disease/tissue/functional effects | Structural effect prediction |
|---|---|---|
| G42D | Galloway–Mowat Syndrome[9] | Very conserved glycine in the ATP binding G-loop of PRPK. May interfere with ATP binding and ATPase or kinase activity. |
| K60Sfs*61 | Galloway–Mowat Syndrome[9] | Large deletion. Kinase domain destroyed. |
| T81R | Galloway–Mowat Syndrome; failed to bind TPRKB[9] | Conserved residue located in helix αC. Not involved in TPRKB binding directly. May distort or destabilize αC, β4, β5, which are required for the binding. May also affect ATPase or kinase activity through αC. |
| R243L | Galloway–Mowat Syndrome[9] | Very conserved, forms a salt bridge with E219. May affect helix αH orientation and thus OSGEP catalytic activity. |
| L174Wfs*4 L174Pfs*23 | Truncation in 9 cases from cBioPortal curated set of non-redundant cancer studies, and 2 cell lines from cancer cell line encyclopedia (2012)[41,42] | Large deletion. Kinase domain destroyed. |
| K238Nfs*2 | Truncation in 3 cases from cBioPortal curated set of non-redundant cancer studies, and 5 cell lines from cancer cell line encyclopedia (2012)[41,42]; Compromised in TPRKB binding (Fig. 6c) | Part of helix αH and the conserved C-terminal tail was deleted. May affect OSGEP catalytic activity. |
| R152* R152P R152Q | 1 truncation and 4 missense mutation from cBioPortal curated set of non-redundant cancer studies[41,42] | Forms a week hydrogen bond with G95 of αC-β4 loop. Not very conserved. Truncation will destroy kinase domain, while mutation effect is unclear. |
| R243C | 2 cases from cBioPortal curated set of non-redundant cancer studies, and 1 cell line from cancer cell line encyclopedia (2012)[41,42] | Very conserved, forms a salt bridge with E219. May affect helix αH orientation and thus OSGEP catalytic activity. |

**Table 2 Disease mutations of human TPRKB.**

| TPRKB mutation | Disease/tissue/functional effects | Structural effect prediction |
|---|---|---|
| L136P | Galloway–Mowat Syndrome[9] | Deeply buried, anchor helix α6 to the core structure. May affect protein structural integrity. |
| Y149C | Galloway–Mowat Syndrome[9] | Deeply buried, hydrophobic core formation, may affect protein structural integrity. |

(b) OSGEP/Kae1 as the catalytic subunit responsible for t6A modification, is extremely conserved across all three domains of life. OSGEP has approximately 35% identity with its ortholog TsaD in *Escherichia coli* (*E. coli*), and 32% identity with *Thermotoga maritima* (*T. maritima*) TsaD. In bacteria, TsaB-TsaD-TsaE serves the same function as the eukaryotic and archaea EKC/KEOPS complex. Beyond the homologous catalytic subunit, the bacterial and eukaryotic complex both contain an ATPase subunit and, in both cases, ATP is sandwiched between the ATPase subunit and the OSGEP/TsaD subunit (Supplementary Fig. S5c). In bacteria, TsaE is a G-loop ATPase, and in eukaryotes, Bud32/PRPK could be converted from a kinase to ATPase in the complex[6]. Another similarity is that bacterial TsaB and eukaryotic Pcc1 both have the ability to dimerize. Moreover, in the bacterial TsaB-TsaD-TsaE complex, a somewhat similar regulatory mechanism exists. TsaE, the G-loop ATPase, binds at the entrance of the TsaD catalytic center, blocking access of tRNA to the site. Notably, a phenylalanine from TsaE (F64) reaches deeply into the pocket (Supplementary Fig. S5d)[32,33]. TsaE is required for a multi-turnover t6A modification reaction, and by hydrolysis of ATP, reset TsaD to a pre-catalytic/active status[32,34]. In our PRPK–OSGEP model, the C-terminal tail also binds and the second arginine of the RGR motif points to the OSGEP catalytic center (Supplementary Fig. 5b, d). The similarity between two t6A modification systems implies that the PRPK C-terminal tail may also block substrate binding and play an inhibitory role. These discussions are hypothetical and future studies are needed to fully address this hypothesis.

Mutations of PRPK and TPRKB are found in human Galloway–Mowat syndrome and in various cancers. We have shown that small molecules targeting PRPK showed promising efficacy in a colon cancer metastasis model and in skin cancer prevention and therapy models[13,14]. Having a human PRPK–TPRKB crystal structure in its liganded form will facilitate more rational drug design in the future.

## Methods

**Protein expression and purification**. The human *RPPK* gene was a gift from Lorenzo A. Pinna[35]. The human *TPRKB* gene was cloned from DLD-1 cells by using standard reverse transcription and PCR technologies. The human *OSGEP* gene was purchased from DNAsu (Tempe, AZ, USA). For expression, all GST-tagged constructs were cloned into the pGEX-6p-1 vector (GE Healthcare; Chicago, IL, USA) and all His-tagged constructs were cloned into the pRSFDuet-1 (Novagen; Madison, WI, USA) vector. All constructs were verified by sequencing (Integrated DNA Technologies, IDT; Coralville, IA, USA).

For structural studies, PRPK and TPRKB were ligated into the multiple cloning site I (MCSI) and MCSII of the pRSFDuet-1 vector, respectively. Thus, PRPK bears an N-terminal His tag, and the PRPK–TPRKB complex was co-expressed in the *E. coli* strain BL21-CodonPlus (DE3). Cells were cultured in Luria–Bertani (LB) medium with 50 μg/ml kanamycin at 37 °C until the OD$_{600}$ of the culture reached 0.8-1.0. Protein expression was induced by 0.25 mM isopropyl-β-D-thiogalactopyranoside (IPTG, GoldBio; St. Louis, MO, USA) for 20 h at 16 °C. The cells were harvested by centrifugation at 4,000 rpm (Thermo Lynx 6000; Waltham, MA, USA). The pellet was resuspended with lysis buffer (20 mM Tris-HCl, pH 8.0, 400 mM NaCl, and 30 mM imidazole) and disrupted by sonication. The lysate was centrifuged at 16,000 rpm for 30 min, and the supernatant fraction was incubated with HisPur Ni-NTA resin (Thermo; Waltham, MA, USA) in batch mode for 2 h. After extensive washing with lysis buffer, the beads were collected into a 10 ml column. Target proteins were eluted with elution buffer (20 mM Tris-HCl, pH 8.0, 100 mM NaCl, and 400 mM imidazole) and then supplemented with 10 mM dithiothreitol (DTT). PRPK–TPRKB proteins were concentrated and loaded onto an anion exchange HiTrap Q HP column (GE Healthcare). Target proteins were eluted with a linear NaCl gradient and further purified using a Superdex 200 Increase 10/300 gel filtration column (GE Healthcare) in buffer containing 20 mM Tris-HCl, pH 8.0, 150 mM NaCl, and 10 mM DTT. Other His-tagged proteins were purified similarly and depending on the applications, only affinity purification and anion exchange chromatography may have been used.

GST-tagged proteins were expressed in the same fashion as PRPK–TPRKB, except that 100 μg/ml ampicillin was used in the LB medium. The harvested cell pellet was resuspended in lysis buffer (20 mM Tris-HCl, pH 8.0, 200 mM NaCl, and 10 mM dithiothreitol (DTT)) and disrupted by sonication. The lysates were cleared by centrifugation at 16,000 rpm for 30 min and applied to glutathione Sepharose 4B

**Table 3 Data collection and refinement statistics (molecular replacement).**

| | PRPK–TPRKB |
|---|---|
| *Data collection* | |
| Space group | *P*2₁ |
| Cell dimensions | |
| *a, b, c* (Å) | 66.86, 77.54, 100.41 |
| α, β, γ (°) | 90, 106.81, 90 |
| Resolution (Å) | 49.36-2.53(2.64-2.53)[a] |
| $R_{merge}$ | 10.7(101.7) |
| $I / \sigma I$ | 11.1(2.0) |
| Completeness (%) | 99.1(99.2) |
| Redundancy | 5.1(5.1) |
| *Refinement* | |
| Resolution (Å) | 2.53 |
| No. reflections | 32629 |
| $R_{work} / R_{free}$ | 0.2114/0.2583 |
| No. atoms | |
| Protein | 6401 |
| Ligand/ion | 90 |
| Water | 65 |
| B-factors | 57.05 |
| Protein | 57.02 |
| Ligand/ion | 67.87 |
| Water | 51.41 |
| R.m.s. deviations | |
| Bond lengths (Å) | 0.007 |
| Bond angles (°) | 0.922 |

[a] Values in parentheses are for highest-resolution shell

resin (GE healthcare). After extensive washing with lysis buffer, the beads were collected into a 10 ml column. On-column cleavage of the GST tag was performed by the addition of homemade PreScission protease and gentle rotation at 4°C overnight. The cleavage buffer consisted of 20 mM Tris-HCl, pH 8.0, 100 mM NaCl, and 10 mM DTT. The target proteins were eluted using the cleavage buffer and concentrated. Anion exchange chromatography (HiTrap Q HP column, GE Healthcare) and gel filtration (Superdex 200 Increase 10/300, GE Healthcare) were used sequentially to further purify the target proteins. Depending on the applications, only affinity purification may have been used and the GST tag may not have been removed.

To purify MBP-tagged OSGEP, the human *OSGEP* gene was cloned into a modified pMal-c2X vector (New England Biolabs; Ipswich, MA, USA). MBP-OSGEP was expressed in the *E. coli* strain BL21-CodonPlus (DE3). When $OD_{600}$ of the cell culture reached 0.8–1.0, protein expression was induced by 0.25 mM IPTG for 20 h at 16°C. The pellet was resuspended with lysis buffer (20 mM Tris-HCl, pH 8.0, 200 mM NaCl, and 10 mM dithiothreitol (DTT)) and disrupted by sonication. The lysates were cleared by centrifugation at 16,000 rpm for 30 min and applied to Amylose resin (New England Biolabs). After extensive washing with lysis buffer, the target proteins were eluted with 20 mM Tris-HCl, pH 8.0, 100 mM NaCl, and 10 mM maltose. Anion exchange chromatography (HiTrap Q HP column, GE Healthcare) was used to separate MBP-OSGEP from the fall-off MBP-tag. Purified proteins were flash-frozen in liquid nitrogen and stored at -80°C.

**Protein crystallization and structure determination**. The purified PRPK–TPRKB protein complex was concentrated to 10 mg/ml, supplemented with 1 mM AMPPNP and 2 mM MgCl₂ (final concentration), and subjected to crystallization screens by the sitting-drop vapor diffusion method at 16°C. To set up trials for crystallization, the protein was mixed with precipitant at a ratio of 1:1 using the Phoenix protein crystallography robot (Art Robbins Instruments; Sunnyvale, CA, USA). Multiple commercial kits were screened, including those from Hampton Research (Aliso Viejo, CA, USA), Jena Bioscience (Jena, Germany), and NeXtal Tubes Protein Complex Suite (Hilden, Germany). Crystals were grown in the reservoir condition of 0.1 M KCl, 0.1 M HEPES pH 7.5, 15% PEG 6000. Crystals were transferred to cryo solutions containing 25% glycerol before being flash-frozen in liquid nitrogen. X-ray diffraction data were collected at The Northeastern Collaborative Access Team (NE-CAT) beamline 24-ID-C at the wavelength of 0.979 Å. Data were processed with the NE-CAT RAPD server, which mainly uses XDS[36]. The human PRPK–TPRKB structure was solved by the molecular replacement method using the program Phaser[37,38]. The human TPRKB structure (PDB ID 3ENP) and *M. jannaschii* Bud32 domain structure (PDB ID 3EN9) were used as search models. Manual model building was performed using

Coot[39] to improve the PRPK structure. The structure was refined with Phenix refine[40], and the final 2.53 Å PRPK–TPRKB structure has a $R_{work}$ and $R_{free}$ of 0.211 and 0.258, respectively. Data scaling, refinement, and validation statistics are listed in Table 3.

**In vitro pulldown**. To examine the interactions between various PRPK mutants with TPRKB or OSGEP, GST-tagged PRPK proteins were first captured onto 20 μl glutathione Sepharose 4B resin from an appropriate amount of BL21 lysates. Then, beads with bound GST-PRPK proteins were incubated with approximately 15 μg purified His-TPRKB or MBP-OSGEP proteins in binding buffer (20 mM Tris-HCl, pH 8.0, 300 mM NaCl, 0.1% Nonidet P-40, 5% glycerol, 2 mM DTT, and 0.4 mM phenylmethanesulfonyl fluoride (PMSF)) for 2 h at 4°C. Beads were washed with binding buffer 4 times, and the bound proteins were detected by Western blotting by using anti-His (Santa Cruz, sc-8036; Santa Cruz, CA, USA) or anti-MBP (Cell Signaling #2396; Danvers, MA, USA).

To test the binding between MTX and PRPK, Methotrexate-agarose suspension from Sigma (Catalogue number M0269) was used. Glutathione-agarose resin was used as control beads. Purified PRPK–TPRKB protein (10 μg) was incubated with 20 μl compound-beads (or control beads) in binding buffer (20 mM Tris-HCl, pH 8.0, 300 mM NaCl, 0.1% Nonidet P-40, 5% glycerol, 2 mM DTT, and 0.4 mM PMSF) for 2 h at 4°C. Beads were washed with binding buffer 4 times, and the bound proteins were detected by Western blotting by using anti-PRPK (Santa Cruz, sc-514703).

**In vitro kinase assay**. For in vitro kinase assay, purified proteins were incubated with 5 μCi [γ-³²P] ATP (PerkinElmer, BLU002A500UC; Waltham, MA, USA) in kinase buffer containing 25 mM HEPES pH 7.4, 2 mM DTT, 0.1 mM Na₃VO₄, and 10 mM MgCl₂ for 1 h at 30°C. Reaction products were separated on SDS-PAGE gel. Gels were dried and exposed to a storage phosphor screen. The protein band with incorporated radioactivity was visualized using a Storm 840 phosphor-imager.

**Computational docking**. Small molecules were docked to the PRPK–TPRKB crystal structure using the docking program Glide 5.9 (Schrödinger LLC; New York, NY, USA). For docking analysis, the ATP binding site based receptor grid was generated, and ligands were prepared by the LigPrep program with default parameters (Schrödinger). Hydrogen atoms were added consistent with a pH of 7.0. Docking was achieved with default parameters in the extra precision (XP) mode[14]. The reported XP GScore of the ATP and MTX is −13.665 and −11.895, respectively.

**Statistics and reproducibility**. Proteins were purified under the same condition and all experiments were conducted in replicates as indicated. The X-ray data collection and refinement statistics were summarized in Table 3.

**Reporting summary**. Further information on research design is available in the Nature Research Reporting Summary linked to this article.

## Data availability
The coordinates and structure factors for the human PRPK–TPRKB–AMPPNP complex were deposited in the Protein Data Bank under accession number: 6WQX. All relevant data are available from the authors upon request.

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

## Acknowledgements
This work was supported by The Hormel Foundation and is based upon research conducted at the Northeastern Collaborative Access Team beamlines, which are funded by the National Institute of General Medical Sciences from the National Institutes of Health (P30 GM124165). The Pilatus 6M detector on 24-ID-C beam line is funded by a NIH-ORIP HEI grant (S10 RR029205). This research used resources of the Advanced Photon Source, a U.S. Department of Energy (DOE) Office of Science User Facility operated for the DOE Office of Science by Argonne National Laboratory under Contract No. DE-AC02-06CH11357. We also want to thank Todd Schuster, manager of the core facility of The Hormel Institute, University of Minnesota, for his efforts in maintaining the crystallization robot and in-house X-ray diffraction system.

## Author contributions
J.L., X.M., and Z.D. designed the research project. J.L. and X.M. performed experiments and analyzed data. S.B. assisted in X-ray diffraction data collection. H.C. performed computational docking. W.M. assisted in performing experiments. J.L. wrote the paper, with input from A.B. and other authors. Z.D. supervised the study.

## Competing interests
The authors declare no competing interests.
