## [Peer Review File · Communications Biology]

Reviewers' comments:

Reviewer #1 (Remarks to the Author):

J Li and coworkers present in this manuscript the crystal structure of the PRPK-TPRKB binary complex in presence of a non-hydrolysable ATP analog (AMPPNP) at 2.5 Å resolution. This complex is part of the 5 subunit human KEOPS complex together with OSGEP, LAGE3 and GON7. Crystal structures of the homologous yeast and archaeal subcomplexes were known before and are very similar to the present human complex. Experiments testing the interactions of interface mutants between PRPK and TPRKB validated the structural data. The structure was further used to construct a 3D model of the complete hKEOPS complex. This model was again tested by the analysis of the interaction of a few mutants at the interface between PRPK and OSGEP.

Although the paper does not provide new insights into the cell biology or mechanistic aspects of the KEOPS complex, it fills in missing structural data. The structure is sufficiently analyzed and clearly presented. In their introduction the authors mention that “small molecules targeting PRPK showed promising efficacy “ and it is surprising that, now they have the structures in hand, they do not try to test the binding of these compounds directly. These studies are not even further alluded to in the discussion. This is regretful since the introduction finishes with the sales-man remark: “As a very promising drug target, its structure will certainly facilitate more rational drug development.”

Reviewer #2 (Remarks to the Author):

The article by Li et al. describes the crystal structure of the human TPRKB-PRPK complex, two subunits of the EKC/KEOPS complex. This study highlights several structural features in PRPK that are related to its kinase/ATPase activity and discusses the similarity and divergence of these features relative to other kinases. This study further describes the details of the interaction between TPRKB and PRPK, highlighting key contact residues. Using their new structure and other existing structures, the authors generate a model for the 5-subunit human KEOPS complex.

Structure of Cgi121-Bud32 complexes, the archaeal and yeast orthologs, have been previously published and are as expected highly similar to that of the human complex. Therefore, the novelty of the reported structure is limited. Almost all of the insights derived from this structure can be inferred by the previously published structures. The authors also do not provide almost any biochemical functional validation for their structural findings.

None the less, the report of the structure should be interesting to researchers in the KEOPS field and potentially to other kinase structural biologists and therefore justifies publication.

Several issues need to be addressed:

1. A central question that is not addressed is what is the role of TPRKB binding to PRPK? Does TPRKB regulate PRPK enzyme activity similar to the archaeal proteins? Does TPRKB regulate the ATP binding activity of PRPK?
2. The authors should provide some biochemical evidence that the residues in PRPK they have highlighted indeed have the expected role in PRPK activity. Specifically, would mutations in G42, K60, D162 and D183 render PRPK inactive? How do these residues compare with those identified in yeast and archaeal Bud32? Is the ATP binding mechanism conserved between the yeast and human proteins?

3. The authors give very little detail of the effect PRPK binding has on TPRKB. Are there any other structural changes induced by PRPK binding, similar to those found in the yeast protein?
4. Description of PRPK as a target for developing small molecule drugs to treat metastatic cancer is somewhat confusing. Evidence points that PRPK has a central role in biosynthesis of an essential tRNA modification. This would argue that inhibiting PRPK activity would be highly toxic for healthy cells.
5. The conserved C terminal tail of PRPK/Bud32 proteins has been discussed previously and has been suggested to have a role in the catalytic cycle of EKC/KEOPS, although the precise role of this element has not yet been disclosed. The authors suggest that this element has a role in inhibiting substrate binding to Kae1/OSGEP and thereby inhibiting t6A modification. Is there any experimental evidence the authors can point to that supports this claim? Would KEOPS supplemented with PRPK/Bud32 truncated in the C terminus bind substrates and be catalytically active? In the absence of experimental data, the authors claims are highly speculative and not convincing.

Response to the Reviewers

Dear Editor and Reviewers,

Thank you for giving us the opportunity to submit a revised manuscript. We appreciate the time and effort that you and the reviewers dedicated to helping improve the manuscript. We have incorporated most of the suggestions made by the reviewers. Those changes are made with “track change” on. Please see below, in blue, for a point-by-point response to the reviewers’ comments and concerns. All line numbers refer to the revised manuscript file with tracked changes.

Reviewers' comments:

Reviewer #1 (Remarks to the Author):

J Li and coworkers present in this manuscript the crystal structure of the PRPK-TPRKB binary complex in presence of a non-hydrolysable ATP analog (AMPPNP) at 2.5 Å resolution. This complex is part of the 5 subunit human KEOPS complex together with OSGEP, LAGE3 and GON7. Crystal structures of the homologous yeast and archaeal subcomplexes were known before and are very similar to the present human complex. Experiments testing the interactions of interface mutants between PRPK and TPRKB validated the structural data. The structure was further used to construct a 3D model of the complete hKEOPS complex. This model was again tested by the analysis of the interaction of a few mutants at the interface between PRPK and OSGEP.

Although the paper does not provide new insights into the cell biology or mechanistic aspects of the KEOPS complex, it fills in missing structural data. The structure is sufficiently analyzed and clearly presented. In their introduction the authors mention that “small molecules targeting PRPK showed promising efficacy “ and it is surprising that, now they have the structures in hand, they do not try to test the binding of these compounds directly. These studies are not even further alluded to in the discussion. This is regretful since the introduction finishes with the sales-man remark: “As a very promising drug target, its structure will certainly facilitate more rational drug development.”

Thank you for pointing out this issue. In the revised manuscript, with the structure available we performed virtual screening of the FDA approved drugs to identify potential PRPK inhibitors. The Schrödinger Glide software was used to perform computational docking. This is similar to what we have done before using a modeled PRPK structure [1]. Unexpectedly, we find methotrexate (MTX) as a potential candidate (Fig. 5a, b; see below). *In vitro* experiments show that MTX could bind and inhibit PRPK (Fig. 5c, d; see below). These data suggest that our crystal structure could be beneficial to virtual screening and further drug development (Line 204-213). These are discussed using a dedicated section now (Line 262-271).

The binding of small molecules (fusidic acid, rocuronium bromide, and betamethasone 17-valerate) to PRPK has been tested in our previous publications using pulldown assays [1, 2]. Therefore, we did not focus on these compounds and chose to seek new ones in the current manuscript.

Reviewer #2 (Remarks to the Author):

The article by Li et al. describes the crystal structure of the human TPRKB-PRPK complex, two subunits of the EKC/KEOPS complex. This study highlights several structural features in PRPK that are related to its kinase/ATPase activity and discusses the similarity and divergence of these features relative to other kinases. This study further describes the details of the interaction between TPRKB and PRPK, highlighting key contact residues. Using their new structure and other existing structures, the authors generate a model for the 5-subunit human KEOPS complex.

Structure of Cgi121-Bud32 complexes, the archaeal and yeast orthologs, have been previously published and are as expected highly similar to that of the human complex. Therefore, the novelty of the reported structure is limited. Almost all of the insights derived from this structure can be inferred by the previously

published structures. The authors also do not provide almost any biochemical functional validation for their structural findings.

None the less, the report of the structure should be interesting to researchers in the KEOPS field and potentially to other kinase structural biologists and therefore justifies publication.

Several issues need to be addressed:

1. A central question that is not addressed is what is the role of TPRKB binding to PRPK? Does TPRKB regulate PRPK enzyme activity similar to the archaeal proteins? Does TPRKB regulate the ATP binding activity of PRPK?

This is an important question. In the revised manuscript, we show that TPRKB could stimulate PRPK autophosphorylation activity (Line 167). Furthermore, OSGEP could inhibit the PRPK-TPRKB autophosphorylation activity (Line 224; Fig. 3d; see below). Both are similar to the yeast and archaeal proteins.

PRPK itself does not express very well and we did not test whether TPRKB regulates PRPK ATP binding experimentally using ITC or other methods. Based on the structure, PRPK N-lobe makes extensive interactions with TPRKB. Particularly, PRPK residue V101 from the regulatory spine forms a hydrogen bond with TPRKB S170. Thus, TPRKB may help maintain the regulatory spine in an assembled states and the PRPK in an active conformation. One possibility is that TPRKB regulates PRPK ATP binding. Another possibility is that TPRKB does not influence PRPK ATP binding, but help position the catalytic elements ready for catalysis. This analysis is now added to the manuscript (Line 168-172).

2. The authors should provide some biochemical evidence that the residues in PRPK they have highlighted indeed have the expected role in PRPK activity. Specifically, would mutations in G42, K60, D162 and D183 render PRPK inactive? How do these residues compare with those identified in yeast and archaeal Bud32? Is the ATP binding mechanism conserved between the yeast and human proteins?

Thank you for the suggestions. We find that K60A, D162N, and D183A mutation significantly decreased the PRPK-TPRKB autophosphorylation activity. Unexpectedly, G42A showed elevated activity. We could only find that in BRAF, replacement of the third glycine of the G-loop with alanine shows a similar stimulatory effect (Fig. 2d; see below).

These residues (G42, K60, D162, and D183) are extremely conserved in general kinase and also in yeast and archaeal Bud32 (Line 133-136; Supplementary Fig. S1a, b; see below). The ATP binding mechanisms are also conserved between the yeast and human proteins (Supplementary Fig. S1a).

3. The authors give very little detail of the effect PRPK binding has on TPRKB. Are there any other structural changes induced by PRPK binding, similar to those found in the yeast protein?

For the human proteins, PRPK binding induces the formation of a short helix ($\alpha 8$) from a segment of loop in TPRKB. No other major structural change could be observed (Supplementary Fig. S3b; see below).

This is different from yeast protein, where major structural change occurs in a segment between helix $\alpha 1$ and strand $\beta 3$. This region is a short helix in the apo Cgi121 structure but transforms into a loop in the Bud32-Cgi121 complex structure (Supplementary Fig. S3c; see below). This analysis is now added to the manuscript (Line 182-187).

4. Description of PRPK as a target for developing small molecule drugs to treat metastatic cancer is somewhat confusing. Evidence points that PRPK has a central role in biosynthesis of an essential tRNA modification. This would argue that inhibiting PRPK activity would be highly toxic for healthy cells.

Thank you for pointing out this issue. In the revised manuscript, we try to make it clear that metastatic cancer has elevated PRPK protein levels and S250 phosphorylation levels. Despite a central role in the synthesis of an essential tRNA modification, the elevated PRPK activity in metastatic cancer compared to normal tissues could be preferentially targeted and toxicity to healthy cells may be minimized. In our studies, mice have been benefited from the treatment using PRPK inhibitors (Line 53-58).

5. The conserved C terminal tail of PRPK/Bud32 proteins has been discussed previously and has been suggested to have a role in the catalytic cycle of EKC/KEOPS, although the precise role of this element has not yet been disclosed. The authors suggest that this element has a role in inhibiting substrate binding to Kae1/OSGEP and thereby inhibiting t6A modification. Is there any experimental evidence the authors can point to that supports this claim? Would KEOPS supplemented with PRPK/Bud32 truncated in the C terminus bind substrates and be catalytically active? In the absence of experimental data, the authors claims are highly speculative and not convincing.

We agree that these are important questions. Due to the focus of this study and technical difficulties faced in a short period of time for revision, we acknowledged that these discussions are hypothetical and future studies are needed to fully address this hypothesis (Line 320).

1. Roh, E., et al., *Targeting PRPK and TOPK for skin cancer prevention and therapy*. *Oncogene*, 2018. **37**(42): p. 5633-5647.
2. Zykova, T., et al., *Targeting PRPK Function Blocks Colon Cancer Metastasis*. *Mol Cancer Ther*, 2018. **17**(5): p. 1101-1113.

REVIEWERS' COMMENTS:

Reviewer #1 (Remarks to the Author):

the authors have made an effort to increase the biological impact of their structural observations by

1) Studying the phosphorylation capacities of a series of mutants of PRPK

These data are of interest to better understand the biochemical function of PRPK

2) Virtual screening of a small molecule data base for potential inhibitors, proposing methotrexate as a potential hit (a bit less convincing)

I suggest accepting the manuscript for publication after a linguistic clean-up

Reviewer #2 (Remarks to the Author):

The authors have satisfactorily addressed our main points.

Two main points with new experimental data and one main point by toning down specific claims in the paper.

I recommend publication in the manuscript's current form